

# The effects of genetic drift and genomic selection on differentiation and local adaptation of the introduced populations of *Aedes albopictus* in southern Russia

Evgenii A. Konorov[1,2], Vyacheslav Yurchenko[3,4], Ivan Patraman[3,5], Alexander Lukashev[3] and Nadezhda Oyun[3,5,6]

[1] Vavilov Institute of General Genetics of Russian Academy of Science, Moscow, Russian Federation
[2] V.M. Gorbatov Federal Research Center for Food Systems of Russian Academy of Sciences, Moscow, Russian Federation
[3] Martsinovsky Institute of Medical Parasitology, Tropical and Vector-Borne Diseases, Sechenov University, Moscow, Russian Federation
[4] Life Science Research Centre, University of Ostrava, Ostrava, Czech Republic
[5] Federal State Budgetary Institution "National Research Centre for Epidemiology and Microbiology named after the Honorary Academician N. F. Gamaleya" of the Ministry of Health of the Russian Federation, Moscow, Russian Federation
[6] Department of Entomology, Biological Faculty, Lomonosov Moscow State University, Moscow, Russian Federation

Corresponding author
Evgenii A. Konorov,
casqy@yandex.ru

## ABSTRACT

**Background:** Asian tiger mosquito *Aedes albopictus* is an arbovirus vector that has spread from its native habitation areal in Southeast Asia throughout North and South Americas, Europe, and Africa. *Ae. albopictus* was first detected in the Southern Federal District of the Russian Federation in the subtropical town of Sochi in 2011. In subsequent years, this species has been described in the continental areas with more severe climate and lower winter temperatures.

**Methods:** Genomic analysis of pooled *Ae. albopictus* samples collected in the mosquito populations in the coastal and continental regions of the Krasnodar Krai was conducted to look for the genetic changes associated with the spread and potential cold adaptation in *Ae. albopictus*.

**Results:** The results of the phylogenetic analysis based on mitochondrial genomes corresponded well with the hypothesis that *Ae. albopictus* haplotype A1a2a1 was introduced into the region from a single source. Population analysis revealed the role of dispersal and genetic drift in the local adaptation of the Asian tiger mosquito. The absence of shared haplotypes between the samples and high fixation indices suggest that gene flow between samples was heavily restricted. Mitochondrial and genomic differentiation together with different distances between dispersal routes, natural and anthropogenic barriers and local effective population size reduction could lead to difficulties in local climatic adaptations due to reduced selection effectiveness. We have found genomic regions with selective sweep patterns which can be considered as having been affected by recent selection events. The genes located in these regions participate in neural protection, lipid conservation, and cuticle formation during diapause. These processes were shown to be important for cold adaptation in the previous transcriptomic and proteomic studies. However, the

population history and relatively low coverage obtained in the present article could have negatively affect sweep detection.

## INTRODUCTION

The Asian tiger mosquito *Aedes albopictus* is a vector of several viruses causing such diseases as yellow fever, dengue fever, and Chikungunya fever (*Akiner et al., 2016*). It is assumed that this mosquito can also transmit the Zika virus (*Gardner, Chen & Sarkar, 2016*). This invasive species has spread from its native habitat in Southeast Asia throughout Africa, North and South Americas, across the Mediterranean Sea to southern Europe, where it was described since the end of the 20th century. Car tires and artificial water reservoirs were presumably the main routes of mosquitoes spread to new territories (*Gratz, 2004*). Climate modeling has suggested that the Asian tiger mosquito may inhabit North Caucasus, the Black and Azov Sea coast, and Kuban River valley (*Benedict et al., 2007*). Regarding the introduction of viruses, the Black sea coast area between Adler and Novorossiisk is under the potential threat. Approximately 200 cases of dengue fever are reported annually in Russia, but no local virus transmission has been recorded so far. It has been pointed out that the risk of arbovirus infection spreading over the South of Russia should not be underestimated (*Ganushkina et al., 2012*; *Sergiev, 2014*; *Yasjukevich et al., 2017*; *Yasjukevich et al., 2013*; *Zabashta, 2016*). Therefore, the Federal Service for Surveillance on Consumer Rights Protection and Human Wellbeing Agency is constantly monitoring the number of introduced mosquitoes and the presence of transmissible pathogens. Currently, both most important arboviral vectors, *Ae. aegypti* and *Ae. albopictus*, occur in the Black Sea region, namely, in Turkey, Georgia, Bulgaria, and in the South of Russia (*Akiner et al., 2016*; *Kotsakiozi et al., 2018*). In contrast to *Ae. albopictus*, *Ae. aegypti* was so far found only in the coastal areas (*Yasjukevich et al., 2017*, *Ganushkina et al., 2016*), which may possibly be explained by this species inability to survive the winter in the inland areas (*Reinhold, Lazzari & Lahondère, 2018*; *Kramer et al., 2020*; *Yasjukevich et al., 2017*).

In a previous phylogenetic analysis of *Ae. albopictus* mosquitoes from the Black Sea region, no significant difference between local populations was found based on COI and ITS2 data, probably because only samples from the subtropical areas in the vicinity of Sochi were used (*Shaikevich et al., 2018*). Whole-genome analyses showed that differences between local populations, found within a limited area (city outskirts), can be as high as between samples collected in the coastal and continental areas. A similar situation was previously reported in Italy and Africa using RADseq (*Kotsakiozi et al., 2017*), where the scenario of in situ divergence after a single or multiple introductions from the same region was posited. In contrast, *Ae. albopictus* mosquito populations in Greece and Albania

showed low level of divergence, which could probably be due to their slower expansion in these countries (*Pichler et al., 2019*).

The *Ae. albopictus* was first found in the Black Sea coastal area in 2011 in Sochi (*Ganushkina et al., 2012*). The previous mosquito monitoring carried out in 2007 did not show the presence of any *Ae. albopictus*, thus the introduction could have taken place between the years 2007 and 2011 (*Ganushkina et al., 2016*). In 2012, *Ae. albopictus* was found along the Black Sea coast (from the Adler to Dzhubga districts) (*Ganushkina et al., 2016*) and 44 km inshore (Krasnaya Polyana) (*Ganushkina et al., 2013*). Over several years, the Asian tiger mosquito has spread along the western coast of the Black Sea to Gelendzhik, Novorossiysk, and Anapa, and migrated inland as far as Maikop, Apsheronsk, and Khadyzhensk (*Zabashta, 2016*; *Fedorova et al., 2018*; *Fedorova et al., 2019b*). In 2018, *Ae. albopictus* populations were found in Krasnodar, on the northern slopes of the Caucasus, in the Adygeya Republic, and the Zakubanskaya plain (*Fedorova et al., 2019b*). Over a few years, the area occupied by *Ae. albopictus* has extended to 260 km along the Black Sea coast and has advanced more than 140 km inland (*Fedorova et al., 2019a*). As reported, *Ae. albopictus* actively spreads in the northwest direction and was found in the villages Yurovka, Slavyansk-on-Kuban, Timashevsk, Korenovsk, and Vostochnoye in the Krasnodar Krai (*Sycheva et al., 2020*).

The Black Sea coast of the North Caucasus is the only Russian territory, where tiger mosquito presence has been documented; however, the processes of *Ae. albopictus* adaptation to the new habitation area with a more continental climate has not yet been investigated. Cold tolerance of *Ae. albopictus* attracted attention in the middle of the 20th century, when this mosquito species started its expansion into the temperate latitudes of North America and Asia. It is considered to be colder-more resistant than another globally prevalent mosquito vector of virus fevers, *Ae. aegypti*. The low threshold temperature for *Ae. aegypti* eggs development is 9 °C (*Tsuda & Takagi, 2001*), while the eggs of *Ae. albopictus* can resist −10 °C (*Thomas et al., 2001*). The recent meta-analysis, however, has challenged the differences in low temperature survival between these two species by demonstrating that *Ae. aegypti* eggs could survive sub-zero temperatures for a short period of time (*Kramer et al., 2020*).

Furthermore, it has been previously observed that *Ae. albopictus* from colder regions with marked seasonality (e.g., United States) differed in their ability to tolerate negative temperatures from the mosquitoes collected in warmer regions (e.g., Malaysia, tropical Africa) (*Hanson & Craig, 1994*). Noteworthy, for *Ae. aegypti* mosquitoes, collected in the same habitats (northern Indiana, USA), no such low temperature resistance was documented (*Kotsakiozi et al., 2017*). At the same time, a recent population genomics study demonstrated a rather high degree of genetic differentiation between these mosquito populations, which suggests that there are some genetic differences between *Ae. albopictus* mosquitoes from different parts of the world, which underlie the differences in cold tolerance (*Kotsakiozi et al., 2017*). Even though the complete genome sequence is available for this species, no whole-genome analysis for specific genes or loci differing in the allele frequency spectrum as a result of natural selection for cold tolerance has been reported so far.

European populations of Ae. albopictus also demonstrated the ability to survive humid continental winters and were more tolerable to overwintering in the same conditions than mosquitoes from tropical zones (*Thomas et al., 2012*; *Tippelt, Werner & Kampen, 2020*). Eight years after the introduction, mosquitoes from these populations have extended their period of activity into the coldest months of the year (*Romi, 2001*). There are many other factors, which affect the spread of Aedes mosquitoes over the areas with temperate climate, including daylight length, precipitation, and atmospheric pressure (*Garzón et al., 2021*; *Benitez et al., 2021*).

There are many ways to survive low temperatures in insects. Among them are metabolic mechanisms, including polyol and ATP synthesis in mitochondria, involvement of heat shock proteins and antioxidants, and specific regulation in response to stress (*Storey & Storey, 2012*). Genes involved in these processes were found to be under selection in the natural populations (*Mallard et al., 2018*, *Božičević et al., 2016*). Some of these mechanisms have already been studied in *Ae. albopictus* (*Kreß et al., 2016*; *Poelchau et al., 2013*), or other mosquito species (*Kayukawa & Ishikawa, 2009*; *Kim et al., 2006*; *Kostál & Simek, 1998*; *Rinehart, Robich & Denlinger, 2006*). Whole-genome sequencing and subsequent population analysis are promising tools for identifying new mechanisms for climate adaptation in insects, but so far these approaches have been used to study cold adaptation mostly in the model species (*Storey & Storey, 2012*; *Xia et al., 2004*). In some cases, nucleotide variants found to be under selection for cold tolerance were localized in non-coding regions, such as introns and cis-regulatory elements (*Pool, Braun & Lack, 2017*; *Wilches, 2014*), thus making it challenging to draw conclusions on their role in cold adaptation given our understanding of cold adaptation physiology. For *Aedes* spp., local adaptation to specific the environment (*Bennett, McMillan & Loaiza, 2021*), cold (*Sherpa, Blum & Després, 2019*), and overwintering (*Medley, Westby & Jenkins, 2019*) was also described.

In the present work, we describe the structure of and genomic changes in *Ae. albopictus* populations in the Krasnodar krai and identified the signatures of potential local adaptations.

## MATERIALS & METHODS

### Sample collection

Mosquito samples were collected between September 17 and 19, 2019 at six locations (Fig. 1) in the Krasnodar Krai (Russian Federation). Three samples (30 mosquitoes in 1_Sochi, 17 in 2_Sochi, and 30 in 3_Sochi) were collected in Sochi, where the climate is humid subtropical without a marked winter. Three samples (42 mosquitoes in 4_Krasnodar, 25 in 5_Krasnodar, and 40 in 6_Krasnodar) were collected in Krasnodar with a more continental climate and below zero winter temperatures. Locations within a city were separated by at least 10 km, while the distance between the two cities is over 100 km. Collection sites were located in city cemeteries (excluding 3_Sochi, which was located at a rabbit farm at the forest edge). Visitors to the cemetery bring many memorial vases, which serve as reservoirs, where immature stages of *Ae. albopictus* develop. Sampling was performed from 6 to 7 pm, when female mosquitoes attack most actively.
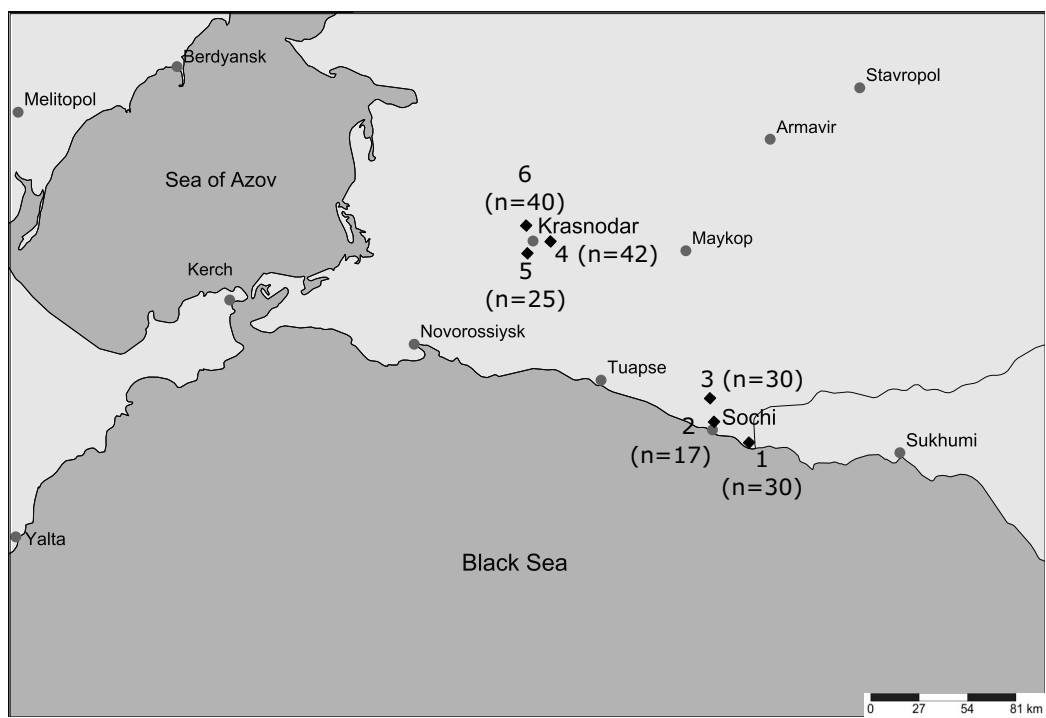

**Figure 1 Location of collection sites.** Location of collection sites designated with diamonds and numbers that correspond to figures and tables below. Generated with SimpleMappr (*Shorthouse, 2010*). Sample size is given in brackets.

Adult female mosquitoes were captured using a mosquito aspirator when they landed and attempted to feed on collectors. After capturing, mosquitoes were identified using the *Gutsevich, Monchadsky & Stackelberg (1970)* Field Guide. Insects were stored in cold 96% ethanol.

## DNA extraction and high-throughput sequencing

Mosquitoes from each collection site were pooled in a 1.5 ml tube containing 180 µl of PBS. Homogenization was performed using TissueLyser II (QIAGEN, Hilden, Germany) at 30 shakes/s for 4 min. DNA was extracted from homogenized samples using the DNeasy Blood & Tissue Kit (QIAGEN, Hilden, Germany) in accordance with the manufacturer's protocol. Short (150 bp) paired-end whole-genome (WGS) reads were obtained by Macrogen Inc. (Seoul, South Korea) using Illumina NovaSeq 6000. Average coverages were 58.4× for Sochi (1.9× per individual), 64.4× for 2_Sochi (3.8× per individual), 57.2× for 3_Sochi (1.5× per individual), 61.4× for 4_Krasnodar (1.5× per individual), 64.3× for 5_Krasnodar (2.6× per individual), and 58.4× for 6_Krasnodar (1.6× per individual). The mean GC-content was 40.28%. Read quality was checked using the FastQC software (*Andrews, 2010*); adaptors and low quality (phred < 20) reads were trimmed using Trimmomatic v. 0.38 (*Bolger, Lohse & Usadel, 2014*). Raw sequence reads for pool-seq samples are accessible by Sequence Read Archive accession numbers SRR13570421 to SRR13570426.
### Read mapping, population genomics, and sweep detection

Reads were aligned in pairs to the *Ae. albopictus* reference genome (Refseq assembly AalbF2 accession GCF_001876365.2) using bbmap (*Bushnell, 2014*). Popoolation (*Kofler et al., 2011*) software was used to estimate population genetics parameters, such as $\pi$, $\theta$, and Tajima's D, in pooled samples using a 1 kb sliding window, and Popoolation2 (*Kofler, Pandey & Schlotterer, 2011*) to estimate Fst and conduct Fisher's exact test to compare differences in the allele frequency spectrum (AFS) between samples (1 kb sliding window, step-size = 1). Genomic regions for which significant (FDR < 0.05, Bonferroni correction) differences in AFS were found between the populations from different cities and not found within Sochi or within Krasnodar were selected.

Single nucleotide polymorphism and indel calling was performed using GATK HaplotypeCaller (*McKenna et al., 2010*). The variance-covariance matrix ($\Omega$) for cluster analysis using the Bayesian hierarchical model was build using the neutral core model (*Coop et al., 2010*) in the BAYPASS software (*Gautier, 2015*). Effective population size was estimated using the poolne_estimate script (*Gautier et al., 2013*).

Selective sweep detection analysis was performed using SweeD (*Pavlidis et al., 2013*), grid = 1,000, for each pooled sample. For scaffolds over 50 Mbp grid = 10000 was used. Scaffolds shorter than 100 kb were excluded from the analysis. Genomic regions affected by sweep were selected based on the Composite likelihood ratio (CLR) greater than a 0.5% quantile in each sample. Gene was considered to be affected by sweep if it was located in a sweep region or if a sweep was located in a putative promotor and transcription binding site region (1 kb upstream). BLAST2GO (*Conesa et al., 2005*) was used for the functional annotation of genes located in sweeps.

### Phylogenetic analysis

Filtered reads were mapped on the reference mitochondrial genome (NC_006817) separately for each sample using bbmap (*Bushnell, 2014*). These mapped reads were used to reconstruct haplotype sequences and frequencies in each sample using RegressHaplo pipeline (*Leviyang et al., 2017*) (max_num_haplotypes = sample size, sig = 0.01, num_trials = 1,000). Mitochondrial genome haplotype sequences of Sochi and Krasnodar samples were aligned using MAFFT (L-INS-i method, *Katoh et al., 2005*) with other available in GenBank *Ae. albopictus* mitochondrial genomes (*Zé-Zé et al., 2020*; *Battaglia et al., 2016*; *Zhang et al., 2016*). We used PartitionFinder (*Lanfear et al., 2012*) to identify a partitioned model of sequence evolution for mitochondrial genome alignment and Mr. Bayes (*Ronquist et al., 2012*) for Bayesian phylogenetic inference (MCMC ngen = 5,000,000). We checked the convergence of the runs by the program Tracer 1.7 (*Rambaut et al., 2018*).

## RESULTS

### Sequencing and read mapping

Whole genome sequencing produced 142.9–161.2 Gb of raw read sequence data, 97% of which were retained after quality trimming. About 71–77% of reads were successfully mapped on the *Ae. albopictus* genome. It resulted in 57.2–64.4× coverage per a pooled

**Table 1 Fixation indices (F$_{st}$) between studied samples.**

|  | 1_Sochi | 2_Sochi | 3_Sochi | 4_Krasnodar | 5_Krasnodar |
|---|---|---|---|---|---|
| 2_Sochi | 0.251 |  |  |  |  |
| 3_Sochi | 0.313 | 0.23 |  |  |  |
| 4_Krasnodar | 0.205 | 0.195 | 0.244 |  |  |
| 5_Krasnodar | 0.136 | 0.323 | 0.386 | 0.272 |  |
| 6_Krasnodar | 0.136 | 0.325 | 0.385 | 0.272 | 0.026 |

sample and 1.5–3.8× coverage per an individual mosquito within a pool. For mitogenome, the coverage was 417–479× per sample.

## Differentiation between populations

Firstly, the suitability and representativity of the obtained sequence data were checked by comparing the sample size and the effective population size. Local populations 1_Sochi and 4_Krasnodar had a three-fold lower effective population size than its actual sample size (9.7 ± 1.7 against 30 individuals and 19.3 ± 3.5 against 42 individuals, respectively), probably due to the isolation that led to inbreeding. Population 3_Sochi showed a less drastic deviation between the effective population and sample sizes (20.3 ± 7.5 vs. 30). Other local populations had effective population size close to the actual sample size, according to the whole-genome allele frequency estimation using poolna_estimate.

Analysis of differentiation between the populations revealed 44.25 million polymorphic sites across the *Ae. albopictus* genome. More than half of them (22.59 million sites) were significantly (FDR < 0.05, Bonferroni correction) different between at least one pair of studied populations. Analysis of fixation indices (Fst) between samples revealed high level of population differentiation (Table 1). Differentiation between the three samples from Sochi was higher than between 1_Sochi and all Krasnodar samples, as well as between 2_Sochi and 4_Krasnodar or 3_Sochi and 4_Krasnodar. Based on the mean Fst values across the genome, only 5_Krasnodar and 6_Krasnodar appeared to be weakly differentiated; the observed differentiation between all other pairs of samples pointed to the high numbers of specific alleles in each population.

Cluster analysis also confirmed strong differentiation between the local populations except for 5_Krasnodar and 6_Krasnodar (Fig. 2). Despite Krasnodar local populations were separated by a similar geographical distance from each other, 4_Krasnodar mosquitoes were genetically as distant from other Krasnodar populations as from 2_Sochi mosquitoes.

## Genomic regions with distinct allele spectrum

To identify genes associated with cold adaptation, we selected genomic regions, which showed significant differentiation between samples from Sochi and Krasnodar, but not between the samples collected within each of the two cities. These loci were filtered from outliers by the Fst value according to the Rosner's test. As a result, we obtained 46

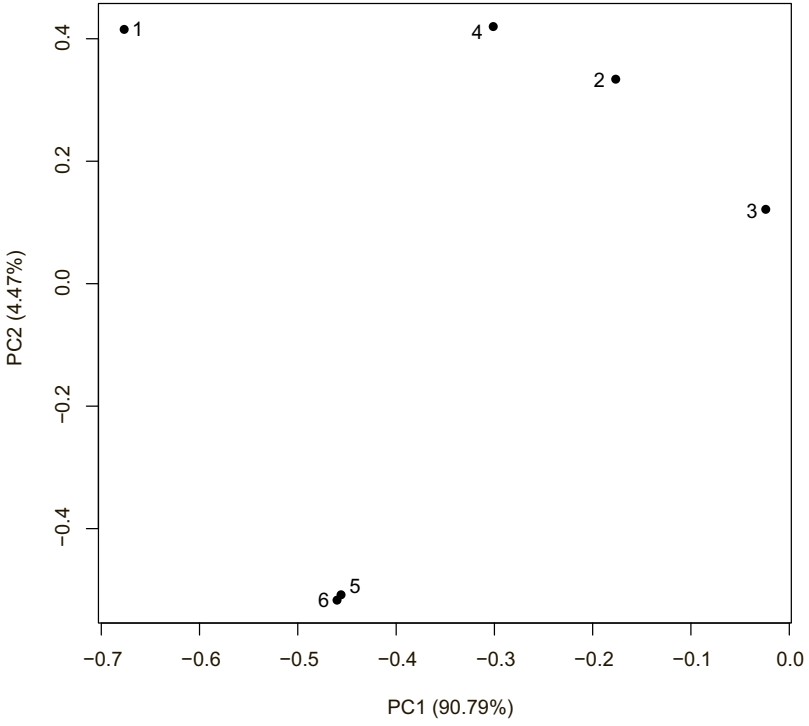

**Figure 2 Clustering analysis of Krasnodar Krai *Ae. albopictus* mosquitoes under Bayesian hierarchical model.** Numbers correspond to collection sites (Fig. 1).

*Ae. albopictus* genomic regions (Table S1) with allele frequencies differing between the Sochi and Krasnodar, but not within the Sochi or Krasnodar populations.

## Selective sweeps

Composite likelihood ratio distribution across the scaffolds with candidate selective sweeps for each local population are presented in Fig. 3. About 30–40% of these regions did not contain genes within the 10 kb range. Genes found in the selective sweep regions are presented in the Table 2.

Three of these genes were involved in fatty acid metabolism (LOC109422645, LOC115263932, LOC115260991, GO:0006633 fatty acid biosynthetic process). Many loci located in selective sweeps were also involved in development, in particular, in neurogenesis and eye development: homeobox protein HMX3, NXF1, DCX, neuronal PAS domain-containing protein 3, PAX-6, and rdgC (GO:0007399 nervous system development). Some of them participated both in the developmental pathways and pathogen response, such as Krüppel-like factors and runt-related transcription factors.

## Phylogenetic analysis and haplotype distribution

Phylogenetic analysis based on the whole mitochondrial genomes showed that despite significant differences in the allele frequency spectrum *Ae. albopictus* populations from the Krasnodar Krai are closely related and share a common origin (Fig. 4). Tiger mosquitoes from Russia formed a separate clade with the posterior probability of 1. Mosquitoes in

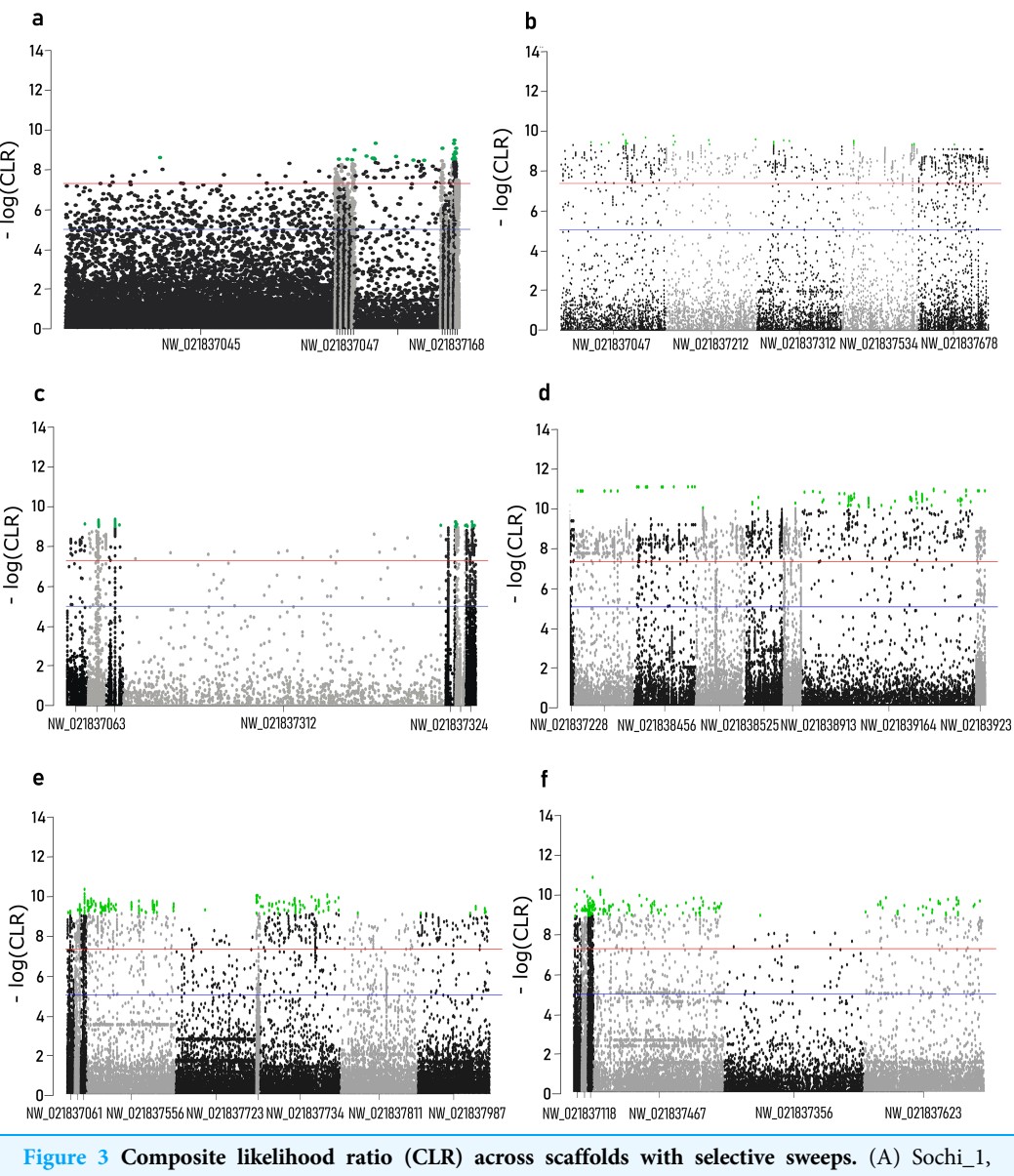

**Figure 3 Composite likelihood ratio (CLR) across scaffolds with selective sweeps.** (A) Sochi_1, (B) Sochi_2, (C) Sochi_3, (D) 4_Krasnodar, (E) 5_Krasnodar, (F) 6_Krasnodar. Green dots indicate genomic regions with CLR greater than a 0.5% quantile in each sample.

this clade appeared to be related to mosquitoes from Italy, Portugal, and Greece, their mitochondrial haplotypes belonging to the haplogroup A1a2. We didn't find any shared mitochondrial haplotype between any pair of analyzed samples. 3_Sochi and 4_Krasnodar populations revealed a single unique mitochondrial haplotype each, while other samples revealed one major (present in 73–80% of the individuals according to PredictHaplo) and two to four less common haplotypes (Table 3). The COI region in all samples of the Sochi and Krasnodar mosquitoes was identical to the one previously found in this region (MG198595, *Shaikevich et al., 2018*).

**Table 2  List of genes located in selected sweep regions.**

| | | Tajima's D | pi |
|---|---|---|---|
| **Sochi_1** | | | |
| LOC115265510 | lncRNA | 0.74 | 0.0027 |
| LOC109422645 | short-chain specific acyl-CoA dehydrogenase | 0.84 | 0.0027 |
| LOC109427049 | frizzled | 0.49 | 0.0022 |
| LOC109422897 | disintegrin and metalloproteinase domain-containing protein 12 | 1,31 | 0.0026 |
| LOC109427779 | hemicentin-2-like | 0,23 | 0.002 |
| LOC109424968 | ras GTPase-activating protein-binding protein 1 | 1.05 | 0.0041 |
| LOC109431083 | chitin deacetylase 1 | 1.15 | 0.0021 |
| LOC109431116 | adenosylhomocysteinase-like 1 | −0.13 | 0.0022 |
| LOC109424976 | titin | 0.11 | 0.0028 |
| **Sochi_2** | | | |
| LOC109425273 | runt-related transcription factor 2-like | 0.22 | 0.0021 |
| LOC109421849 | segmentation protein Runt-like | −0.09 | 0.0017 |
| LOC109412943 | tyrosine-protein phosphatase non-receptor type 23-like | 0.07 | 0.0018 |
| LOC109417087 | sodium/potassium/calcium exchanger 3-like | 0.08 | 0.0012 |
| LOC109418656 | Krueppel-like factor 5 | 0.85 | 0.0027 |
| **Sochi_3** | | | |
| LOC109427647 | lncRNA | 1.03 | 0.0047 |
| **Krasnodar_4** | | | |
| LOC115260991 | fatty acid synthase-like | 0.24 | 0.0034 |
| LOC109432289 | single-stranded DNA-binding protein 3 | 1.17 | 0.004 |
| LOC115267667 | larval/pupal cuticle protein H1C-like - | 0.57 | 0.0034 |
| LOC109408817 | potassium channel subfamily T member 1 | 1.05 | 0.0044 |
| LOC109409075 | calcyphosin-like protein | 0.52 | 0.0032 |
| LOC109622526 | facilitated trehalose transporter Tret1-like | 0.28 | 0.0024 |
| LOC109622532 | PDZ domain-containing protein GIPC3-like | 0.37 | 0.0025 |
| LOC109409257 | syntaxin-1A | 0.39 | 0.0033 |
| **Krasnodar_5** | | | |
| LOC109397006 | paired box protein Pax-6-like | 0.22 | 0.0035 |
| LOC115262968 | homeobox protein HMX3-like | 0.15 | 0.0017 |
| LOC115262971 | serine/threonine-protein phosphatase rdgC-like | 0.53 | 0.0035 |
| LOC115262973 | serine/threonine-protein phosphatase rdgC-like | 0.36 | 0.005 |
| LOC115263932 | elongation of very long chain fatty acids protein AAEL008004-like | 0.94 | 0.0049 |
| LOC109426122 | serine/threonine-protein phosphatase PP1-beta catalytic subunit | 0.74 | 0.0048 |
| LOC115263945 | putative ATP-dependent RNA helicase DHX57 | 1.35 | 0.0071 |
| LOC109426144 | nuclear RNA export factor 1 | 0.45 | 0.0032 |
| LOC115263947 | open rectifier potassium channel protein 1-like | 0.67 | 0.0045 |
| **Krasnodar_6** | | | |
| LOC109421031 | protein transport protein Sec61 subunit alpha-like | 0.85 | 0,0067 |
| LOC115262466 | deoxynucleotidyltransferase terminal-interacting protein 2-like | 0.73 | 0,004 |
| LOC109427703 | serine/threonine-protein kinase GL21140 | 0.69 | 0,0043 |

| Table 2 (continued) | | | |
|---|---|---|---|
| | | Tajima's D | pi |
| LOC109427739 | peroxidase | 0.39 | 0.0045 |
| LOC115262467 | uncharacterized connector enhancer of ksr | −0.12 | 0.0038 |
| LOC115262469 | ras-related and estrogen-regulated growth inhibitor-like protein | 0.46 | 0.005 |
| LOC109427854 | dnaJ homolog subfamily C member 7 | 0.29 | 0.0054 |
| LOC115262471 | small nucleolar RNA U3 | 0.00 | 0.001 |
| LOC109428139 | acidic amino acid decarboxylase GADL1-like | 0.16 | 0.0036 |
| LOC115263426 | neuronal PAS domain-containing protein 3-like | 0.58 | 0.0059 |

The haplotypes identified in the 1_Sochi, 2_Sochi and 5_Krasnodar samples formed a distinct clade. In contrast, the rare 6_Krasnodar haplotypes (Krasnodar_6_H4 and Krasnodar_6_H3) showed more basal localization on the phylogenetic tree relative to all other Sochi and Krasnodar haplotypes (Fig. 4).

## DISCUSSION

### Experimental design limitations and coverage effects

We have selected *Ae. albopictus* populations from Sochi and Krasnodar because of distinct differences in winter temperatures between these two regions. However, it should be noted that *Ae. albopictus* can also be found in other areas of the Krasnodar Krai, where climatic conditions are different from those in both chosen locations. In Novorossiysk and Tuapse, located in the northern part of the Black sea coast, the average daily temperature does not differ dramatically from that in Sochi, but average precipitation is lower and annual average wind speed is higher (gismeteo.ru). It may cause differences in selective pressure and patterns of local adaptation. It is also expected that there is a gene flow between these places, which can limit the efficiency of selection. There is no data confirming that Sochi was the immediate source of mosquito expansion to Krasnodar, thus selective pressure imposed by some other factors apart from low winter temperatures could have shaped the variation patterns.

To accurately detect selection sweeps it is necessary to reduce the demographic history effects, since the bottlenecks and founder effects are hardly distinguishable from selective sweeps when working with small samples (*Jensen et al., 2007*). Thus, the 2_Sochi ($n = 17$) and 5_Krasnodar ($n = 25$) samples may be considered too small to efficiently detect sweeps with relatively small selection coefficients. To reduce demographical effects, we had planned to find selection sweeps in the same genomic regions in all three Sochi populations compared to all three Krasnodar populations to be sure that selection search results are reliable. We found no genomic regions of this kind in the Sochi and Krasnodar populations, although certain genes under selection in different samples belong to the same pathways or share the same function.

Furthermore, the sweep detection using low coverage pool-seq data may not be accurate, because of the wrong base calling or analyzing just one chromosome of a diploid

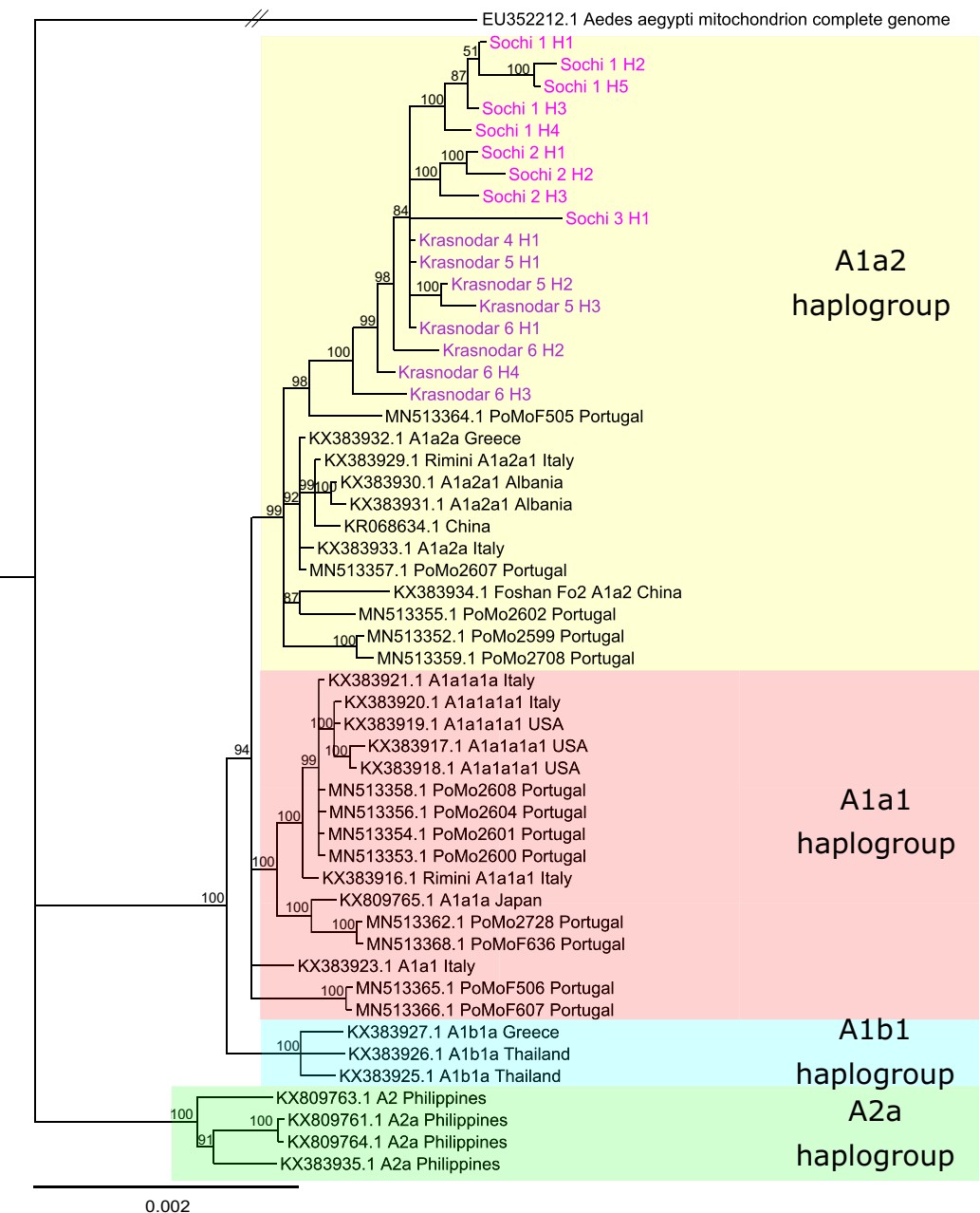

**Figure 4 Bayesian phylogeny based on *Aedes albopictus* mitochondrial genomes.** Alignment length 14,367 bp. The substitution model was set for each gene and intergenic region separately using PartitionFinder. Sochi and Krasnodar haplotypes marked pink and purple, respectively

organism (*Pavlidis et al., 2013*). For a reliable detection of 30% allele spectrum differences between two populations both coverage and pool-size of 50 are needed (*Kofler et al., 2011*). Most of our samples passed this threshold, but two (2_Sochi and 5_Krasnodar) were slightly deficient in pool sizes. Of note, for a pairwise Fst analysis, the results obtained with even lower coverage data did not drastically differ from the classical estimates

| Table 3 Mitochodrial haplotype frequencies for each sample. | |
| --- | --- |
| Haplotype | Frequency (%) |
| 1_Sochi | |
| Sochi_1_H1 | 73.8 |
| Sochi_1_H2 | 8.7 |
| Sochi_1_H3 | 8 |
| Sochi_1_H4 | 5 |
| Sochi_1_H5 | 4 |
| 2_Sochi | |
| Sochi_2_H1 | 80 |
| Sochi_2_H2 | 11.1 |
| Sochi_2_H3 | 6.2 |
| 3_Sochi | |
| Sochi_3_H1 | 100 |
| 4_Krasnodar | |
| Krasnodar_4_H1 | 100 |
| 5_Krasnodar | |
| Krasnodar_5_H1 | 73 |
| Krasnodar_5_H2 | 21.4 |
| Krasnodar_5_H3 | 5.6 |
| 6_Krasnodar | |
| Krasnodar_6_H1 | 76.4 |
| Krasnodar_6_H2 | 12.4 |
| Krasnodar_6_H3 | 3.8 |
| Krasnodar_6_H4 | 2.9 |

Note:
No common haplotypes between samples was found.

(*Hivert et al., 2018*), so we may assume that the quality of our data was sufficient for differentiation analysis.

## Tiger mosquito dispersal and bottlenecks

In a previous phylogenetic analysis of *Ae. albopictus* mosquitoes from the Black Sea region, no significant differentiation between local populations was found using COI and ITS2 data, probably because only samples from subtropical areas near Sochi were studied (*Shaikevich et al., 2018*). It should be noted that COI marker lacks the resolution to discriminate Krasnodar krai *Ae. albopictus* populations, as only two haplotypes were found previously (*Shaikevich et al., 2018*). In agreement with that, no variations were found in the COI region in the current work.

Genetic differentiation study using PoolSeq revealed that *Ae. albopictus* populations from Russia belong to the same cluster and clade (Fig. 4), suggesting that the populations within the region diverged as a result of in situ differentiation after a single or multiple introduction(s) from the same geographical region. Similar results were reported previously for *Ae. albopictus* invasion regions in Italy, Africa, and South America

(*Kotsakiozi et al., 2017*). While clear differentiation between different geographical populations in Italy and Africa was demonstrated, its level (fixation indices) was about two times lower than that observed between Sochi and Krasnodar. In contrast, *Ae. albopictus* mosquitoes from Greece and Albania showed low level of divergence from each other, probably due to the lower rates of this mosquito expansion into these countries (*Pichler et al., 2019*). However, in another study, a different scenario for *Ae. albopictus* invasion in Italy was proposed. According to *Manni et al. (2017)*, Italian *Ae. albopictus* populations were a result of multiple introductions and admixtures with the populations from Reunion and the United States. Somewhat controversial results were obtained in the two subsequent ddRADseq studies of Italian *Ae. albopictus* mosquitoes—no recent gene flow between Italy and Reunion (*Sherpa et al., 2018*), but admixture between the Northern Italy and China mosquitoes in Central Italy was demonstrated (*Sherpa et al., 2019*). The *Ae. albopictus* mosquitoes from Russia form a separate clade on the mtDNA phylogenetic tree. This implies a single or several introductions from the same geographic location. It is plausible that unrelated mtDNA haplotypes have been independently introduced to other seaports in the Krasnodar Krai, but it is unlikely that they have spread to the continental parts of the region, since dominant mtDNA haplotypes detected in Krasnodar differed non-significantly from the ones detected in Sochi (Fig. 4). This means that tiger mosquitoes from southern Russia descended from a group of closely related females.

Rather unexpectedly, we found no mitogenome haplotypes shared between the studied samples (Table 3). This could possibly be due to a sampling bias, but, taking into account fixation indices between populations (Table 1), it may rather be an indication of the highly restricted gene flow between populations. Average Fst in this case were higher than those observed between the populations from other invasion regions such as Europe, continental Africa (*Kotsakiozi et al., 2017*; *Pichler et al., 2019*), and Brazil (*Multini et al., 2019*).

It is noteworthy that certain populations from different cities appeared to be more closely related to each other than the population within the same city (Table 1, Fig. 2). It can be accounted for by multiple independent introductions of mosquitoes from the coast to the inland areas, multiple introductions from other areas, and very intensive diversification of mosquito populations in Sochi. The latter case may be exemplified by the 3_Sochi and 1_Sochi populations exhibiting high level of divergence. Local geographic conditions favored diversification, since these mosquito populations were localized to the rural areas separated by foothills, Sochi center, and other water basins. Additionally, 1_Sochi population was collected closer to the Adler seaport, while 2_Sochi and 3_Sochi were collected closer to the Sochi seaport. Thus, the differences between these populations could be a result of separate introductions (although from the same general source, as the mitochondrial sequences suggest) without subsequent merge of the established populations. The 4_Krasnodar population exemplifies how populations may become isolated even within a generally endemic area. It was sampled on a graveyard surrounded by urban areas. The effective population size analysis suggested that the population was affected by inbreeding, while fixation indices indicated that it was significantly

different from other populations. In contrast, in the regions with a dense network of waterbodies gene flow increases (*Medley, Jenkins & Hoffman, 2015*). The similarity between the 5_Krasnodar and 6_Krasnodar populations could be explained by sharing the same water body, namely a Kuban river bend with multiple piers, which may contain small artificial reservoirs, and the proximity to the A146 highway leading to Novorossiysk. At the same time, the 4_Krasnodar population was found at a certain distance from the river transport system and closer to the M4 highway, thus allowing different introduction histories for these populations. This hypothesis fits well into the observation that Sochi mosquitoes are less closely related to each other compared that those from Krasnodar (Table 1). This may be due to the fact that Sochi populations are separated by foothills and rivers or the independent introduction of mosquitoes, sharing similar mitochondrial genomes, but different allele frequencies. Previously, lower genetic differentiation was shown for *Aedes* mosquitoes inhabiting large permanent water storages compared to those inhabiting the temporary ones (*Paupy et al., 2004*). In the urban areas, different pattern and density of oviposition sites (*Huber et al., 2002*) and availability of human-operated transportation means were a more important differentiation factor than geographical distance and natural barriers (*Medley, Jenkins & Hoffman, 2015*).

The *Ae. albopictus* spread across the Krasnodar Krai likely included multiple parallel transmissions of mosquitoes between the cities, as judged by the relatively low levels of differentiation observed between the samples from different cities compared to within-city differentiation. Very high fixation indices and absence of shared haplotypes between samples are compatible with severe bottleneck effects and, possibly, difficulties in local climatic adaptation due to reduced selection effectiveness.

## Cold adaptation

There was no uniform pattern of genomic regions that differed in the allele frequency spectrum between the Sochi and Krasnodar samples detected. Some of the genes in the selective sweep regions identified in different local populations, however, shared common functions, such as fatty acid metabolism (GO:0006633) or neural system development (GO:0007399).

Genes involved in fatty acid metabolism are also often involved in cold adaptation (cold tolerance) in insects (*Kayukawa et al., 2007*; *Michaud & Denlinger, 2006*; *Yang et al., 2018*). In 1_Sochi, 4_Krasnodar, and 5_Krasnodar populations, selective sweeps were found in the genomic regions containing the genes encoding short-chain specific acyl-CoA dehydrogenase, proteins participating in the elongation of very-long-chain fatty acids, and fatty acid synthase-like proteins. Previously, these genes were found to be involved in egg desiccation resistance and lipid conservation during diapause (*Reynolds et al., 2012*; *Urbanski et al., 2010*). As the egg is the only stage in the tiger mosquito development cycle that can survive sub-zero temperatures (*Kreß et al., 2016*), it is expected that selection would act upon genes functioning in diapausing eggs. In addition to the genes involved in fatty acid metabolism, there were other loci affected by selection that could be involved in adaptation at the diapause stage. For example, chitin deacetylase and larval/pupal

cuticle protein H1C may change the cuticle composition at the critical developmental stages (*Chen et al., 2010*; *Dunning, 2013*).

Selection acting on serine/threonine-protein phosphatases rdgC and Pax-6, as well as other proteins responsible for brain development, is consistent with the previous findings which linked cold adaptation to the thermotaxis rhodopsin pathway (*Parker et al., 2015*; *Pavlides, Pavlides & Tammariello, 2011*). In *Drosophila* spp., genes participating in neuronal development were under positive selection in cold-resistant species (*Parker et al., 2018*). Neural protection during diapausing stages may be crucial for further brain development.

Although field data indicate that *Ae. albopictus* expansion into the Krasnodar Krai occurred very recently (*Ganushkina et al., 2016*; *Fedorova et al., 2018*, *2019b*), the complexity of local populations exceeded our expectations. While we achieved sufficient genome coverage, the lack of common haplotypes among populations, relatively high genetic distances, and potentially multiple introduction routes suggest that the number of sampling locations needs to be increased. Due to the technical and logistical limitations, the present study could not cover the whole dispersal history and cold adaptation of *Ae. albopictus* in southern Russia. In the coastal area, there were other locations, where *Ae. albopictus* mosquitoes were recorded. These locations have different climatic conditions to adapt to, such as the rainfall amount and wind speed. In the case of the samples used in the present work, different local landscapes and different distances to the seaports and other spread routes could result in high divergence between the Sochi populations and could have significantly affected our ability to identify genes associated with cold adaptation. In other conditions, 2–3 samples from a geographic area could be sufficient to analyze pool-seq data (*Asgharian et al., 2015*; *Konorov, 2018*), but in this study, the sampling coverage was obviously not exhaustive, likely due to background drift effects.

## CONCLUSIONS

Population genetics and phylogenetic analysis of *Ae. albopictus* pooled samples from Sochi and Krasnodar revealed the possible scenarios of Asian tiger mosquito introduction and spread across southern Russia. The monophyly of all the mitochondrial genome haplotypes in the studied samples from Sochi and Krasnodar suggested a single introduction or, possibly, multiple transmissions from the same location in Europe. The absence of shared haplotypes between the samples and high fixation indices suggested that gene flow between samples is heavily restricted. We suggest that different distances between the spread routes, as well as natural and anthropogenic barriers led to distinct gene flow patterns, local effective population size reduction and difficulties in local climatic adaptations due to reduced selection effectiveness, and may account for the failure to find any uniform adaptation patterns in the present work. Genomic regions with a selective sweep, which could be the signatures of recent selection events, contain genes involved in neural protection, lipid conservation, and cuticle formation during diapause. These processes were found to be important for cold adaptation in previous transcriptomic and

proteomic studies, but linking them unambiguously to the cold adaptation in the Krasnodar Krai may require additional studies.

### Funding

Collection of mosquitoes, sample preparation, and bioinformatics were supported by the Russian Science Foundation (RSF) (project No. 19-75-00091, "Genetic analysis of vector capacity and cold adaptation in Asian tiger mosquito *Aedes albopictus* from lab-contained and introduced natural populations from Russian Federation"). Sequencing of mosquito genomes was provided by the European Regional Funds (project No. CZ.02.1.01/16_019/ 0000759). The funders had no role in study design, data collection and analysis, decision to publish, or preparation of the manuscript.

### Grant Disclosures

The following grant information was disclosed by the authors:
Russian Science Foundation (RSF): 19-75-00091.
European Regional Funds: CZ.02.1.01/16_019/0000759.

### Competing Interests

The authors declare that they have no competing interests.

### Author Contributions

- Evgenii A. Konorov conceived and designed the experiments, performed the experiments, analyzed the data, prepared figures and/or tables, authored or reviewed drafts of the paper, and approved the final draft.
- Vyacheslav Yurchenko performed the experiments, authored or reviewed drafts of the paper, and approved the final draft.
- Ivan Patraman performed the experiments, authored or reviewed drafts of the paper, and approved the final draft.
- Alexander Lukashev conceived and designed the experiments, authored or reviewed drafts of the paper, and approved the final draft.
- Nadezhda Oyun conceived and designed the experiments, performed the experiments, prepared figures and/or tables, authored or reviewed drafts of the paper, and approved the final draft.

### Data Availability

Raw sequence reads for pool-seq samples are available at Sequence Read Archive: SRR13570421 to SRR13570426.

### Supplemental Information

Supplemental information for this article can be found online at http://dx.doi.org/10.7717/ peerj.11776#supplemental-information.

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
