# Peer review of "The effects of genetic drift and genomic selection on differentiation and local adaptation of the introduced populations of Aedes albopictus in southern Russia"

_PeerJ, doi:10.7717/peerj.11776_

## Round 0.1 · original submission · Major Revisions

Dear Dr. Konorov and colleagues:

Thanks for submitting your manuscript to PeerJ. I have now received three independent reviews of your work, and as you will see, one reviewer recommended rejection, while another suggested a major revision. I am affording you the option of revising your manuscript according to all three reviews but understand that your resubmission may be sent to at least one new reviewer for a fresh assessment (unless the reviewer recommending rejection is willing to re-review).

In general, the reviewers wish to see a much-improved manuscript as far as presentation and clarity. There are a lot of suggestions that will help here. There also appears to be literature that needs citing. Both reviewers 2 and 3 feel that conclusions are reached that are not supported by the data. Finally, the conclusions should be assessed in light of all of the appropriate studies in the field.

There are many minor problems pointed out by the reviewers, and you will need to address all of these and expect a thorough review of your revised manuscript by these same reviewers. I agree with the concerns of the reviewers, and thus feel that their suggestions should be adequately addressed before moving forward.

Therefore, I am recommending that you revise your manuscript, accordingly, taking into account all of the issues raised by the reviewers.

I look forward to seeing your revision, and thanks again for submitting your work to PeerJ.

Good luck with your revision,

-joe

Reviewer 1 ·

Basic reporting

Quality is sufficient. Substance is always clear, correct, and unambiguous.

The authors did summarize the main research question and key findings. The authors identified the most important literature on the topic and explain how the study relates to this previously published research.

The arcticlestructure is mainly professional (details see below).

The figures and tables are clear and readable. The figure and table captions are complete and accurate. Are the axes are labeled correctly. The presentation is appropriate for the type of data being presented. The figures and tables support the findings.

The study was conform to ethical guidelines.
The experiments are innovative and the method used seems to be appropriate for addressing the research question.

The statistical analysis is adequate

Authors collected and interpreted data accurately. The authors followed best practices e.g. by using state of the art data logging technology.

The results support most of the conclusions. In one point the conclusions overreach. The authors did discuss some limitations of the study.

Experimental design

The paper is originally primary research within the Scop of the journal.

The research question is well definde and important. I Support the journal's Editors in publishing this article after revisions in order to fill a gap of knowledge.

The research was performed to a high standart in technic an ethics.

The methods discribed were sufficient detailed and all nessecary information is available.

Validity of the findings

All data have been provided and analyse are adhequate.

Some Parts in the Conclusion needs to be better linked to references and the hypothesis needs to be explained more robust.

the limitations of the experimental end analytical design needs to be discussed more openly.

Additional comments

Keywords:
The name of the species is already part of the title of the manuscript and therefore redundant for search engines.

Abstract and introduction:
The authors did summarize the main research question and key findings. The authors identified the most important literature on the topic and explain how the study relates to this previously published research.
Line 83: Recent analyses of cold threshold reveal that winter temperature may not be the limiting factor of expansion of Ae. aegypti in northern regions: see Kramer et al. Parasites Vectors (2020) 13:178 https://doi.org/10.1186/s13071-020-04054-w.
Line 88: Reference for this is needed. I Suggest: Thomas et al. Parasites & Vectors 2012,5:100 http://www.parasitesandvectors.com/content/5/1/10 or Tippelt et al (2019) PLoSONE 14(7):e0219553.https://doi.org/10.1371/journal.pone.0219553
Line 95 and 261: Authors Name is Kreß not Krebs.

Material and methods:
The study was conform to ethical guidelines.
The experiments are innovative and the method used seems to be appropriate for addressing the research question.

Statistical analysis:
The statistical analysis is adequate

Results:
Authors collected and interpreted data accurately. The authors followed best practices e.g. by using state of the art data logging technology.
Line 172ff: hypothesis of close relation due do a shared waterbody is not a result rather a part of the discussion section. Furthermore, this hypothesis seems a little shaky to me. Please add more references e.g. to similar findings in order to support the hypothesis that the waterbody is the main causation rather than correlation.

Discussion, conclusion:
The results support most of the conclusions. In one point the conclusions overreach. The authors did discuss some limitations of the study.
Line 202: Give reference for this statement (Shaikevich als well?)
Line 199-224: A lot of this is (important) literature, but belongs in the introduction part.
Line 224-227: Please be more specific in comparison of the investigated Russian population to the Italian e.g. by comparing specific numbers or dimensions. Please integrate your finings better in the references and data available: discuss them. These two statements came a bit out of the blue.
Line 229: Authors give thee explanations for their finding. However, you focus on the one that is the most unlikely in my eyes. In order to have a diversification, there had to be a dispersal around the city in the past and than a separation of the populations. However, I cant see any reason, why this dispersal should not take place constantly. The foothill where always there and didn’t appeare after the dispersal event. Therefore, I think it is more likely, that there is a constant introduction of Ae. albopictus eggs from the warmer marine sites in the mainland by a cargo. Those eggs perhaps even have a better hatching rate compared to frost damaged ones in the mainland. And by doing so, you have a constant gene flow into these city sites, eliminating other alleles that are dispersed by free flying individuals. Picture it like actively pumping genes in some city sites!
Therefore I think, it is of utmost importance Authors give their theory of diversification way more argumentative background, because it requires more assumptions than the first two explanations.
Line 233: Ae. albopictus are container breeding species. They do not prefer open water bodies as breeding sites. But perhaps there is a river cargo line the main cause?
Line 249: Discuss the limitation of your experimental and analytical design.
Line 281: Authors my explain, why they exlude the possibility of an epigenetic adaptation to winters and by doing so giving the manuscript an bigger impact on the competeing hypothesis of rapid adaptation on cold hardines.
Line 282: Conclusion is more sound to me than discussion.

Figures and tables:
The figures and tables are clear and readable. The figure and table captions are complete and accurate. Are the axes are labeled correctly. The presentation is appropriate for the type of data being presented. The figures and tables support the findings.

Writing quality & clarity
Quality is sufficient. Substance is always clear, correct, and unambiguous.

Reviewer 2 ·

Basic reporting

The ms 57483 compares the genetic structure of six pools samples of Aedes albopictus collected in two localities of the Black Sea Region. Genetic differentiation among samples is studied to identify genomic regions with selective sweeps, which are assumed to be a signature of cold adaptation.
The ms seems rushed, with overreaching conclusions that are not supported by the data. Figures are not numbered correctly in the text.

The introduction section needs more detail on the geography of the Black Sea region and its arboviral vectors. For instance, there are reports of Aedes aegypti in the Black Sea region (https://pubmed.ncbi.nlm.nih.gov/29980229/). Do authors know if Ae. aegypti and Ae. albopictus are sympatric in any areas of the Black sea?

Thermo tolerance is quite a complex phenotype and while there are several examples of genetic traits related to thermal adaptation (see 10.1146/annurev-genet-110711-155511), this has not been studied in mosquitoes yet. Also, the invasion process of Ae. albopictus was quick and occurred through bottlenecks and multiple introductions, often from different places, intermingling, so saying that genetic changes observed among Ae. albopictus populations are related to cold adaptation is really a strong and not supported assumption.

Experimental design

The sampling strategy is not clear. There are no details on whether mosquitoes were sampled as larvae or adults, the season of the sampling which could explain some of the observed differences, the pooling of the mosquitoes. the number of mosquitoes that was pooled influences has impact on the depth of coverage. It is unclear if the declared 20X coverage is over all the pool or calculated on single mosquitoes within the pool (20X would be acceptable in this case). Accession number of a repository where the raw sequencing data have been deposited is lacking.

Validity of the findings

The ms has several shortcomings in the method section (no description of the sampling strategy, of the pooling size, lack of details on read coverage and quality control of the sequencing data) that prevents to fully understand the impact of the results.
Additionally, assuming that all detected differentiation is related to cold adaptation is a big assumption. There may be many other explanations for the observed differences, especially because no overlapping genomic regions were observed, not even whin samples from Sochi vs Krasnodar Krai.

Additional comments

Thermo tolerance in mosquitoes is an important research subject, especially considering the invasive capacity of Ae. albopictus. However, I do not think this ms is about thermo tolerance in Ae. albopictus, rather it describes genetic differentiation of mosquitoes from two localities of the Black Sea. As such I do not find the title appropriate.

Detailed comments below:

1) there are reports of Aedes aegypti in the Black Sea region (https://pubmed.ncbi.nlm.nih.gov/29980229/). Do authors know if Ae. aegypti and Ae. albopictus are sympatric in any areas of the Black sea?

2) Lines 72-75. Thermo tolerance is quite a complex phenotype and while there are several examples of genetic traits related to thermal adaptation (see 10.1146/annurev-genet-110711-155511), this has not been studied in mosquitoes yet. Also, the invasion process of Ae. albopictus was quick and occurred through bottlenecks and multiple introductions, often from different places, intermingling, so saying that genetic changes observed among Ae. albopictus populations are related to cold adaptation is really a strong and not supported assumption.

3) Line 88-89. I think the statement “European populations of Ae. albopictus also demonstrated adaptation to lower temperatures than in tropical zones” is unclear and deceitful. Ae. albopictus populations in Europe are invasive populations, which derive from mosquitoes from different parts of the world, including tropical regions. No controlled experiment has been done yet comparing their fitness in tropical vs temperate conditions to support ideas of adaptation.

4) Line 91-95. There is a bit of confusion here between the phenotype associated with overwintering (diapause) and the molecular mechanisms underlying this process, which have been studied, but not completely fully understood yet. Cited references include studies related to identifying the molecular underpinnings of diapause.

5) Lines 100-101. This work describes the genetic structure of Ae. albopictus populations in the Black Sea. Assuming that the described genetic structure is causative of the invasion ability and/or results of adaptation is a long shot, not supported by the data.

6) Line 104-109. Please specify the season in which the mosquitoes were collected and whether mosquitoes were collected as adults or larvae. Can the three sampling sites withing Sochi and/or Krasnodar Krai be considered replicate of the same population? Where the collected at the same time/year?

7) Line 110-117. Pool size? Sex of mosquitoes in the pool? Morphological identification of Ae. albopictus was performed? This is especially important in case of sympatry with Ae. aegypti. Sample coverage of 20X means 20X for each mosquito in the pool or average for the pool? It is quite important to understand the pool size to estimate if the depth of coverage is enough.

8) Lines 121. The genome assembly is called AalbF2.

9) Lines 145-177. To me the first paragraph of the results should describe the overall experiment design, summarise the sequencing data and provide quality control measurements for the whole-genome sequencing data. Also details of the sampling locations are presented as possible explanations for the observed genetic differences, but I think a more unbiased way would be to first describe the sampling sites in detail, than present the genetic data and try to connect the two in the discussion section.

10) Line 155. Should be figure 2. Please mark the samples from this study.

11) Lines 185-190. To me this analysis shows differentiation between mosquitoes from Sochi and Krasnodar, not necessarily related to cold adaptation.

12) Line 186-187. Please specify what these regions with no genes are.

13) Line 188 is unclear, please re-phrase.

14) Line 225. How does high genetic diverge among samples connect with a quick spread? Authors analysed only mosquitoes from two locations, I would be caution in making generalised statements. Also, if spread was quick, isn’t it more probable to observe low genetic differentiation? High genetic differentiation would be expected in presence of strong bottle necks and limited gene flow among populations and/or introductions from multiple sources.

Reviewer 3 ·

Basic reporting

Clear and unambiguous, professional English used throughout? No. English is non-standard or confusing in several areas of the manuscript.
L34: "dissemination" is used in non-standard way, consider revising to "spread".
L40: what is meant by "selection effect restriction"?
L38 and L54: "transmissions" is used in a non-standard way, consider revising to "migrations" or "dispersal events".
L169: by "special" do you mean "spatial"?
L242: what is meant by "selection interbreeding level"?
L276 and L290: "apparently" is not used correctly in either case. You need to make a statement and back it up with specific evidence.
L277-280: This sentence is confusing, I do not know what is meant.

Literature references, sufficient field background/context provided? No. The references and background is far from sufficient. The motivation for the study is left simply as "it hasn't been done yet" without adequate context to convince the reader this is a novel and important study.
L72: I am confident there are more recent references, please update.
L78: Consider including the barriers to completing full-genome study of adaptation.
L97-99: This statement is inaccurate. There are many studies outside of model species that have used whole-genome sequences to study cold adaptation. Please update the references, and change this part of the manuscript to provide sufficient background to motivate this study.

Professional article structure, figures, tables... and raw data shared? No. The structure does not conform to standard practice. I suggest you make subheadings that match in the methods and results so that the reader can easily find all matching methods and results. As it is there are significant omissions in both sections, and it is very unclear which results are being discussed in the discussion. The discussion should include specific evidence for all interpretation that points to the specific table/figure that displays the result.
L137: There is no section in the methods that describes the window size searched for genes close to the SNPs. There is also no section explaining the gene identification or GO term analysis.
L150: sample size is not the same as census size.
L157-158: Is this the only section reporting on the phylogenetic analysis of the mitochondrial genomes? I do not see adequate reporting of this part of the study.
L187: why 10 kb? Typically this decision is based on average recombination rates, or specific recombination rates for the genomic region of interest if available. There is no mention or justification of this 10 kb range in the methods. Please add this.
L197: I am expecting phylogenetic analysis here (to match the order of the methods), but do not see any results on this part of the study.
L210-223: This belongs in the introduction (if anywhere).
L225: "high divergence between samples" refers to which analysis/result/table/fig?
L228-229: "closely related" refers to which analysis/result/table/fig?
L236: "as the mitochondrial sequences suggest" refers to which analysis/result/table/fig? What is the specific evidence for this?
L252: "partly shared common functions" refers to which analysis/result/table/fig? This is very vague, please quantify what is "partly" and point us to the correct result to look at.
L285-287: "Monophyletic origin..." This was not stated in the results, and we have not seen clear treatment of it previously in the manuscript. Everything in the conclusions should be familiar to the reader.

Self-contained with relevant results to hypotheses. Yes. I do think that the results are relevant to the questions posed, the writing is what needs to change.

Experimental design

Original primary research within Aims and Scope of the journal. Yes.

Research question well defined, relevant & meaningful. It is stated how research fills an identified knowledge gap. Yes, but writing needs to be improved to make it clear.

Rigorous investigation performed to a high technical & ethical standard. Yes, but the context with credit given to previous research in this area is still needed.

Methods described with sufficient detail & information to replicate. No. See comments above in section 1.

Validity of the findings

No comment.

Additional comments

I think this paper is focused on an interesting topic, and the scope of the study and methods are appropriate. However, the writing is not clear, and the structure of the paper is confusing and insufficient. The introduction needs to be greatly expanded to provide context of the study, with clear and comprehensive references to previous studies in this area of research. There are many other studies in this area in insects (outside of model systems) and the authors need to read these studies and summarize them to put this study into context.

To improve organization and ensure you have presented methods and results comprehensively, I suggest you make subheadings that match in the methods and results so that the reader can easily find all matching methods and results. As is, there are significant omissions in both methods and results sections, and it is very unclear which results are being discussed in the discussion.

It is also my opinion that it is a requirement that the discussion include specific evidence for all interpretation that points to the specific table/figure that displays the result in discussion.

Finally, the conclusion should only deal with results that have been clearly presented and discussed, it should not introduce entirely new results or concepts.

---

## Round 0.2 · accepted · Accept

Dear Dr. Konorov and colleagues:

Thanks for revising your manuscript based on the concerns raised by the reviewers. I now believe that your manuscript is suitable for publication. Congratulations! I look forward to seeing this work in print, and I anticipate it being an important resource for groups studying the population genetics of Aedes albopictus in southern Russia. Thanks again for choosing PeerJ to publish such important work.

Best,

-joe